# A Holographic-Type Model in the Description of Polymer–Drug Delivery Processes

**DOI:** 10.3390/ph17040541

**Published:** 2024-04-22

**Authors:** Irina Nica, Constantin Volovat, Diana Boboc, Ovidiu Popa, Lacramioara Ochiuz, Decebal Vasincu, Vlad Ghizdovat, Maricel Agop, Cristian Constantin Volovat, Corina Lupascu Ursulescu, Cristian Virgil Lungulescu, Simona Ruxandra Volovat

**Affiliations:** 1Department of Odontology-Periodontology, Fixed Prosthesis, “Grigore T. Popa” University of Medicine and Pharmacy, 700115 Iasi, Romania; irinanica76@yahoo.com; 2Department of Medical Oncology-Radiotherapy, “Grigore T. Popa” University of Medicine and Pharmacy, 16 University Str, 700115 Iasi, Romania; simonavolovat@gmail.com; 3Department of Emergency Medicine, “Grigore T. Popa” University of Medicine and Pharmacy, 700115 Iasi, Romania; ovidiu.popa@umfiasi.ro; 4Faculty of Pharmacy, “Grigore T. Popa” University of Medicine and Pharmacy, 700115 Iasi, Romania; lacramioara.ochiuz@umfiasi.ro; 5Department of Biophysics, Faculty of Dental Medicine, “Grigore T. Popa” University of Medicine and Pharmacy, 700115 Iasi, Romania; decebal.vasincu@umfiasi.ro; 6Department of Biophysics and Medical Physics, “Grigore T. Popa” University of Medicine and Pharmacy, 700115 Iasi, Romania; vlad.ghizdovat@umfiasi.ro; 7Department of Physics, “Gheorghe Asachi” Technical University of Iasi, 700050 Iasi, Romania; magop@tuiasi.ro; 8Romanian Scientists Academy, 050094 Bucharest, Romania; 9Department of Radiology, “Grigore T. Popa” University of Medicine and Pharmacy, 700115 Iasi, Romania; cristian.volovat@yahoo.com (C.C.V.); corina.ursulescu@gmail.com (C.L.U.); 10Department of Oncology, University of Medicine and Pharmacy of Craiova, 200349 Craiova, Romania; cristilungulescu@yahoo.com

**Keywords:** drug delivery, fractal/multifractal curves, Schrödinger and Madelung scenarios, multifractal diffusion laws, nanomedicine

## Abstract

A unitary model of drug release dynamics is proposed, assuming that the polymer–drug system can be assimilated into a multifractal mathematical object. Then, we made a description of drug release dynamics that implies, via Scale Relativity Theory, the functionality of continuous and undifferentiable curves (fractal or multifractal curves), possibly leading to holographic-like behaviors. At such a conjuncture, the Schrödinger and Madelung multifractal scenarios become compatible: in the Schrödinger multifractal scenario, various modes of drug release can be “mimicked” (via period doubling, damped oscillations, modulated and “chaotic” regimes), while the Madelung multifractal scenario involves multifractal diffusion laws (Fickian and non-Fickian diffusions). In conclusion, we propose a unitary model for describing release dynamics in polymer–drug systems. In the model proposed, the polymer–drug dynamics can be described by employing the Scale Relativity Theory in the monofractal case or also in the multifractal one.

## 1. Introduction

The search for new approaches to drug delivery systems, as well as new ways of delivering drugs, are emerging areas of cancer research. This search involves diversified scientific methodologies to achieve substantial progress regarding therapeutic index and bioavailability, in particular, drug delivery [1,2,3,4].

Drug delivery systems incorporate at least one of the traditional drug delivery approaches with engineering technologies. Therefore, it is feasible to precisely indicate some characteristics of the delivery, like the location where a substance was injected into the body and/or the rate of distribution.

Innovative medication delivery approaches integrate an appropriate selection of biodegradable and bioabsorbable polymers. These elements, such as hydrogels like poly(lactic acid) and poly(glycolic acid) and their copolymers, are employed in developing the delivery structure of these systems [5,6].

The mainstreaming of nanotechnology demonstrated therapeutic efficacy regarding the development of cancer, encouraging the accelerated progress of targeted therapy and combination drug therapy [4,7]. As a result, nano drug-delivery systems (NDDS) are now known as an emerging field of inquiry due to advantages such as their optimal loading, selective delivery, and controlled release. Thus, silicon-based nanomaterials [8,9], polymers [10], liposomes, and metal NPs [11] are used to deliver anti-cancer pharmaceuticals to tumor cells. Polymerized nanomaterials represent promising candidates useful in treating cancer strategies due to their versatile adhesion capacity and flawless biological compatibility [12,13].

### 1.1. Polymer-Based Drug Delivery Systems

Polymer-based therapy delivery systems employ polymers as vehicles that control the release of pharmaceuticals, having the critical objectives of increasing patient adherence and the effectiveness of the drug while minimizing adverse reactions through directed and consistent delivery [14].

Taking into consideration the kind of polymer used, polymeric DDSs are commonly identified into the following categories: natural, synthetic, and hybrid polymers [15,16,17,18].

Natural polymers, which are derived from polysaccharides or proteins, can be efficiently used to improve the results of some therapeutic applications such as cancer therapy, tissue donation, and personalized medicine [19]. Thus, elements thoroughly investigated for this objective, such as chitosan, hyaluronic acid, silk fibroin, and collagen, have proven their efficacy. It can be stated that their potential results both from their high compatibility with the human body and also from their monitored destruction through enzymatic processing. Knowing their potential, it is vital to regulate their action and efficacy by enhancing their capacity to specifically target stimuli-induced behavior [20].

Synthetic polymers such as polyethylene glycol (PEG), poly(lactic-co-glycolic acid) (PLGA), polyvinyl alcohol (PVA) bioabsorbable polymers, biodegradable polymers, dendritic polymers, and poly(ethyleneimine) are produced in well-known and controlled conditions.

Metal–organic frameworks (MOFs) have been introduced as delivery systems with significant potential, owing to their intrinsic biological degradation that limits long-term harmful effects, despite their structure. Nevertheless, there are still some issues about the safety of their usage, given that a few of them fail to stay constant in physiological settings [21,22].

However, it is desirable to employ this deficiency of stability to boost certain features, including the solubility of specific compounds, by amplifying their harmful effects and thereby enhancing the body’s response to the provided therapy [23,24,25].

There are a great variety of beneficial aspects to employing these polymers, resulting not only from their biological characteristics like significant compatibility and degradation but also from the chemical structures that potentiate their mechanical flexibility [26,27]. In controlled DDS formulations, synthetic polymers have greater importance than biopolymers because they have a significant potential for the design and modification of their physicochemical structure [28].

As a mixture of the properties of both the above-mentioned types of polymers, hybrid polymers may provide improvements in biocompatibility, mechanical features, and drug dispersal [27].

Polymer-based DDSs are developed to deliver substances in numerous ways by regulating their discharge through diffusion, erosion, and stimuli reaction [14,29].

As a result, the drug’s release can be constrained by its distinctive diffusion coefficient and gradient of concentration between the polymer matrix and the environment, polymer degradation features, or changes in external constants such as pH, temperature, and light [6,30,31].

A commonly utilized method for investigating polymers, drugs, and their conduct is employing molecular modeling approaches, such as reproducing interaction between different fundamental elements—MD simulations. These simulations are used in the polymer-based DDS literature, given their ability to improve the drug–polymer pair dynamics [32,33].

### 1.2. Polymeric Systems as Nanocarriers

NPs are defined as dispersions of particles or solid particles with small sizes, between 10 and 1000 nm [34]. The classification of nanoparticles included liposomes [35], polymers [36], organic compounds (carbon nanotubes) [37], inorganic NPs, and metallic NPs (silver, gold, or magnetic NPs, etc.) [34].

By employing them, research highlights their ability to reinforce certain pharmacological properties of pharmaceuticals once they are carried and released, playing major roles in prolonging the duration of their half-lives in vivo and preserving a consistent distribution in targeted tissues due to physiological interaction with reticuloendothelial system cells, specifically phagocytic cells. In that regard, it may be claimed that several physicochemical qualities, such as size, shape, and surface characteristics, contribute to their goal [38,39,40].

The essential characteristics of drug delivery systems are the dimension of the particle and size distribution of NPs particles [41]. These two elements determine the in vivo grafting time, distribution, potential harmful effects, and specificity of NP systems [42], but they also impact the loading, release, and stability of drugs in NPs. Particle size greatly affects the release of a drug, considering that smaller particles have a greater exposure area and that the drug is commonly attached next to the surface, permitting an accelerated release [43].

Despite the positive aspects mentioned, small NPs are rapidly destroyed in the liver and spleen, making their utilization in practice somewhat inefficient.

Also, large surface areas of particles may not circulate properly in small vessels. Consequently, dimensions and material nature represent vital characteristics that may impact the effectiveness of NPs related to cancer treatment [44,45].

The biodistribution of NPs presents two interrelated difficulties that are correlated with NPs pharmacokinetics: the distribution of NPs in undesirable locations within the body and reaching a maximum concentration in target locations, both of which are correlated with misguidance or blockage of NPs in areas that have barrier properties [46].

Another obstacle is the protein corona phenomenon, demonstrated in the protein corona at the NP surface, at the contact between NP material and biological components in the body [47]. The composition of the protein corona depends on the physicochemical properties of the polymeric NPs and the actual circulation time. Also, the barrier capacity may interact with the drug’s efforts to maintain a proper concentration at the target site, influencing the efficacy of the treatment response.

These barriers include restrictive mononuclear phagocytic system (MPS) action, cellular internalization, limited vascular circulation, non-specific distribution, pressure variations, avoidance of endocrine and lysosomal mechanisms, and drug excretion pumps [48].

Thus, NPs must be designed considering these biological barriers that prevent their clinical efficacity in cancer treatment. Nanoparticles will be ineffective in combating cancer unless nanomethods address the biological hurdles they confront once inside an organism.

To avoid limitations related to biological obstacles as well as those related to biodistribution, ligands and receptor-targeting proteins are used in the strategic direct route administration of therapeutic NPs as well as for transportation involving cells. In the circulation environment, the surface has a key position in the NPs lifetime in correlation with macrophage uptake throughout the entire course of action [49]. As a result, a hydrophilic surface may be more resistant to macrophage action.

Polymeric nanoparticles, as a part of nanosystems, are developed by a polymerization reaction of different monomer units and are obtained from synthetic, semi-synthetic, or natural polymers [20,50]. NPs show a great diversity of their properties, and for this reason, they are attractive as multifunctional nanocarriers within drug delivery systems (DDS) [51].

Polymeric NPs occur as nanospheres, nanocapsules, or drug conjugates (Figure 1) [20,50], being broadly grouped into five types, namely capsules, micelles, nanogels, dendrimers, and mixed NPs with porous cores [52].

Despite their structural variety, polymeric NPs improve tumor response to the administered drug and reduce side effects. Some of them can also target monoclonal antibodies or antibody fragments, peptides, aptamers, and small molecules conjugated to the material forming the shell with great specificity [52,53,54,55,56,57,58,59,60]. The process depends on different variables defining the ligand, which have the possibility to accentuate receptor internalization and modify drug biodistribution.

### 1.3. Polymer–Drug Delivery Systems (DDS)

Drug delivery systems (DDS) have a major role in the successful and targeted administration of pharmaceuticals. Despite this, there are challenges related to their design and optimization, given the complex interaction between pharmaceutical composition and delivery methods. Considering this, the evolution of DDS has improved the therapeutic efficacy of drugs by influencing their properties determined by their interaction with the human body [61,62]. Whatever the form of administration used for DDS, the drug’s blood level is maintained by the body’s own mechanisms of absorption, distribution, metabolism, and excretion. By analyzing these mechanisms, we can anticipate ineffective levels in target tissues and unwanted implications for tissues [63].

Nanotechnologies featuring DDS have lately drawn considerable attention due to their capacity to manipulate drug solubility while simultaneously safeguarding it against degradation or clearance from the body [30,64,65,66,67].

Accordingly, DDS also facilitates the delivery of precisely targeted medications to particular cells, including cancer cells, using the determinant characteristics of the tumoral environment or surface molecules [68,69].

Conventional drug delivery systems (capsules, tablets, syrup, etc.) are characterized by a high removal rate from the organism and are associated with a fast drug metabolism. Therefore, the therapeutic dose of the drug is not constant when reported to the therapeutic window, with an exponential decrease in plasma level, followed by a short time frame, resulting in a sub-therapeutic response. Another delivery system is a controlled delivery system, with constant drug levels within the therapeutic window and optimal clinical impact for a longer duration, targeting specificity, and improved bioavailability [70]. The known prolonged resistance of the drug is also a consequence of the particular enzymatic reaction within the metabolism, and it also involves a modified schedule of administration. In addition, the biological environment is less exposed to the detrimental impact of the drug, which has important benefits for patient clinical status and compliance.

The controlled delivery systems can be classified into (1). Dissolution-Controlled Drug Delivery Systems, where the polymeric membranes or matrices are slowly dissolute; (2). Diffusion-Controlled Drug Delivery Systems, where the drugs are captured and, after that, liberated by diffusing through polymeric membrane matrices that are unable to dissolve in water; (3). Water Penetration-Controlled Drug Delivery Systems that include Osmotic-Controlled Drug Delivery Systems and Swelling-Controlled Drug Delivery Systems; and (4). Chemically Controlled Drug Delivery Systems, which have their chemical architecture personalized after contact with human body biochemical conditions [2].

DDS is, thus, becoming imperative for the precision approach, as it allows the development of drugs that address the genetic characteristics and disease traits of an individual while minimizing the toxicity risk and revolutionizing the way drugs are administered and delivered to patients [71,72,73].

Therefore, recognizing their possible limitations in clinical practice is crucial. Given the documented instability of several compounds in the environment that imitate the human organism, it is plausible that significant cytotoxicity may occur when their rate of absorption is uncontrolled [74,75]. Simultaneously, susceptibility to specific environmental variables might influence the quality of the anti-tumor therapeutic response [76,77]. Also, a better understanding of the target cells is required in order to benefit from more extensive therapy while reducing the hazardous exposure of the remaining cells [78,79].

### 1.4. Polymer–Drug Release Dynamics

The standard models used to describe the polymer–drug dynamics rely on various semi-empirical laws (Higuchi, Korsmeyer–Peppas, Hixson–Crowell, etc.), their diversity evidently being correlated with the complexity of the drug release mechanisms [31]. As of yet, there is no singular model that can completely explain the drug release dynamics and, implicitly, the mechanisms behind these dynamics.

In such a context, a possible way to develop a unitary model for describing drug pharmacokinetics is according to the prediction that any polymer–drug system may be integrated into a multifractal mathematical concept [80,81], both from a structural and functional perspective. Then, in drug release processes, fractal/multifractal curves can be associated with the polymer–drug structural unit dynamics. Since these curves display the self-similarity property (a single unit represents the entire system, and the whole reflects the part, miming holographic-type behaviors), we can state that, through holographic-type models, we can describe various drug release dynamics. Recent papers support our approach [82,83,84].

The mathematical model that employs an overview of dynamics through continuous and non-differentiable curves can be found in the Scale Relativity Theory, either in the fractal dimension, $D_{F}=2$, like in the monofractal model [84], or in various unchanged and arbitrary fractal dimensions, like in the multifractal model [85,86,87]. In either of these two models, we can distinguish two scenarios for describing dynamics: the multifractal Schrödinger scenario (based on Schrödinger equations at various scale resolutions) and the multifractal Madelung scenario (based on hydrodynamic-type equations at various scale resolutions). These two scenarios are not disjointed but complementary.

In the present paper, a unitary model for drug release dynamics is presented, assuming that the polymer–drug system can be assimilated into a multifractal mathematical object.

## 2. Results

In this paper, we made a description of drug release dynamics that implies, via the Scale Relativity Theory, the functionality of continuous and undifferentiable curves (fractal or multifractal curves), possibly leading to holographic-like behaviors for such dynamics. Such a holographic description of drug release dynamics (in which the whole reflects the part and vice versa, i.e., the self-similarity of fractal/multifractal curves) implies the two aforementioned scenarios: a multifractal Schrödinger scenario, which “mimes” several drug release modes (through period doubling, damped oscillations, modulated and “chaotic” regimes), and a Madelung scenario, which implies various multifractal diffusions.

## 3. Discussions

Our analysis can be a novel study method for drug release dynamics. Because the holographic-type behaviors of drug release processes can be assimilated with deep learning methods [88], our model could connect the discrepancy between conception and the real implementation of holographic imaging methods in the study of drug release dynamics [89,90]. In this context, theoretical simulations employing our model can be correlated with data-driven approaches, which make use of holographic image reconstruction methods for monitoring drug release efficiency [91,92].

The limitation of our model (related to time consumption, accuracy, predictability, etc.) must be correlated with the experimental drug release curves, a situation in which the scale resolution and, implicitly, the fractal dimension for the release curves must be determined. Let us note that, after each release process, the resulting polymer structure is different than the previous (before release) structure, so that, by assimilating the polymer–drug system to a multifractal, we can take these successive changes into consideration (a specific resolution scale is compatible with a specific polymer structure, at different release times). The standard imaging methods employed for determining the polymer structure after release can be correlated with the drug release curves. These release curves can be tuned with the resolution scale from our model and implicitly with the release dynamics presented in Figure 2, Figure 3, Figure 4 and Figure 5.

## 4. Methods

### 4.1. A Brief Recall of the Multifractal Schrödinger and Madelung Scenarios

Let it be considered the multifractal Schrödinger equation [93,94,95]:(1)$$2\lambda^{2}(dt{)}^{\left[\frac{4}{f\left(\alpha \right)}\right]-2}\partial^{l}\partial_{l}\Psi +i\lambda (dt{)}^{\left[\frac{2}{f\left(\alpha \right)}\right]-1}\partial_{t}\Psi =0$$
where
$$\partial^{l}\partial_{l}=\frac{\partial^{2}}{{{\partial x}_{l}}^{2}}\;,\;\partial_{t}=\frac{\partial }{\partial t}\;,\;l=1,2,3$$

In Equation (1), $x_{l}$ are the multifractal space coordinates, t is a non-multifractal time coordinate, Ψ is the state function, λ is a constant correlated to the multifractal-non-multifractal scale transition, dt is the scale resolution, $f\left(\alpha \right)$ is the singularity spectrum of order α with $\alpha =\alpha \left(D_{F}\right)$, and $D_{F}$ the fractal dimension of the motion curves [82,83]. For other details referring to the values of the previously mentioned factors, please see [85,86,87].

Thus, through Equation (1), the multifractal Schrödinger scenario in the statement of polymer–drug dynamics can be substantiated.

Now, for Ψ in the form (the Madelung substitution):(2)$$\Psi =\sqrt{\rho }e^{is},$$
where $\sqrt{\rho }$ is the amplitude, and s is the phase, the complex velocity field [31,78,79,80,81,82,83,84,85]
(3)$${\hat{V}}^{i}=-2i\lambda (dt{)}^{\left[\frac{2}{f\left(\alpha \right)}\right]-1}\partial^{i}\ln\Psi$$
takes the explicit form:(4)$${\hat{V}}^{i}=2\lambda (dt{)}^{\left[\frac{2}{f\left(\alpha \right)}\right]-1}\partial^{i}s-i\lambda (dt{)}^{\left[\frac{2}{f\left(\alpha \right)}\right]-1}\partial^{i}\ln\rho$$
Relation (4) implies the real velocity fields:(5)$$V_{D}^{i}=2\lambda (dt{)}^{\left[\frac{2}{f\left(\alpha \right)}\right]-1}\partial^{i}s$$
(6)$$V_{F}^{i}=\lambda (dt{)}^{\left[\frac{2}{f\left(\alpha \right)}\right]-1}\partial^{i}\ln\rho .$$
where $V_{D}^{i}$ is the differentiable velocity field, i.e., the velocity at differentiable resolution scale, and $V_{F}^{i}$ is the non-differentiable velocity field, i.e., the velocity at non-differentiable resolution scale.

In the following, by relations (2), (5), and (6), and applying calculus techniques from [85,86,87], Formula (1) is reduced to the multifractal hydrodynamic equations:(7)$$\partial_{t}V_{D}^{i}+V_{D}^{l}\partial_{l}V_{D}^{i}=-\partial^{i}Q$$
(8)$$\partial_{t}\rho +\partial_{l}\left(\rho V_{D}^{l}\right)=0$$
with Q the specific multifractal potential:(9)$$Q=-2\lambda^{2}(dt{)}^{\left[\frac{4}{f\left(\alpha \right)}\right]-2}\frac{\partial^{l}\partial_{l}\sqrt{\rho }}{\sqrt{\rho }}=-V_{F}^{i}V_{F}^{i}-\frac{1}{2}\lambda (dt{)}^{\left[\frac{2}{f\left(\alpha \right)}\right]-1}\partial_{l}V_{F}^{l}.$$

Equation (7) correlates to the specific multifractal momentum conservation law. Equation (8) is related to the multifractal state’s density conservation law. Moreover, the specific multifractal potential (9) implies the specific multifractal force:(10)$$F^{i}=-\partial^{i}Q=-2\lambda^{2}(dt{)}^{\left[\frac{4}{f\left(\alpha \right)}\right]-2}\partial^{i}\frac{\partial^{l}\partial_{l}\sqrt{\rho }}{\sqrt{\rho }}$$
which is an assessment of the multifractality of motion curves.

Thus, through Equations (7) and (8), the multifractal Madelung circumstances in the description of polymer–drug kinetics can be substantiated.

### 4.2. Some Implications of the Multifractal Schrödinger Scenario

Controlled drug release is a complex phenomenon, subject to a series of factors. Among these factors, we can highlight the following: (a) formulation type (hydrogel formulation type, liposomes formulation type, etc., and their chemical composition, implicitly); (b) the pH of the medium (we remind the fact that for an acidic or basic pH, the formulation behaves in a different manner); (c) drug solubility; and (d) various external stimuli (temperature, external fields, etc.). However, no matter these mentioned factors, controlled drug release is reducible, from a general point of view, to interaction forces between drug and matrix: First, the loosely bound drug is released (surface release), and after, the strongly bound drug is released (in-depth release). Thus, various release mechanisms are explicated: diffusion, swelling, erosion, etc. Moreover, their functionality order depends, among other things, on the formulation type (e.g., in the case of hydrogels, swelling is followed by erosion).

Mathematically, the above-mentioned factors can be quantified by analyzing both local and global dynamics for structural units of the polymer–drug system assimilated to a complex system. To this purpose, the correlation of dynamics (at any scale resolution) implies special operational procedures: (i) synchronization and simultaneity of dynamics at any scale resolution through a multifractal Schrödinger scenario and (ii) diffusion classes (Fickian diffusion, non-Fickian diffusion, etc.) at various scale resolutions through a multifractal Madelung scenario. In such a context, the selection of one of the previously mentioned factors can be possible only by adequately choosing the scale resolution (and implicitly, the choosing of a fractal dimension for the drug release curves). For example, the Fickian diffusion can be achieved on the basis of fractalization through Markov-type stochasticization (Markov-type stochastic processes) for a fractal dimension, $D_{F}\to 2$, while the non-Fickian diffusion can be obtained on the basis of fractalization through non-Markov-type stochasticization (non-Markov-type stochastic processes for a fractal dimension $D_{F}<2$. Also, the influence of pH, crucial for drug release, can be described through a specific selection of the resolution scale (and, intrinsically, of the fractal dimension for the drug release curves) [59,87].

### 4.3. Synchronizations in Polymer–Drug Dynamics through a Hidden Symmetry

In the one-dimensional stationary case, Equation (1) uses the shape:(11)$$\frac{d^{2}\Psi }{dx^{2}}+k_{0}^{2}\Psi =0$$
with
(12)$$k_{0}^{2}=\frac{E}{2m_{0}\lambda^{2}{\left(dt\right)}^{\left[\frac{4}{f\left(\alpha \right)}\right]-2}}$$
where E is the multifractal energy of the polymer–drug structural unit and $m_{0}$ is the rest mass of the polymer–drug structural unit.

The result of Equation (11) takes the explicit form
(13)$$\Psi \left(x\right)=ze^{i\left(k_{0}x+\theta \right)}+\bar{z}e^{-i\left(k_{0}x+\theta \right)}$$

In relation (13), z is the complex amplitude, $\bar{z}$ is the complex conjugate of z, and θ is a phase.

Consequently, z, $\bar{z}$, and θ mark any of the polymer–drug-established components that possess the identical $k_{0}$.

Equation (11) already features symmetry achieved through the incorporation of a homographic group. In fact, the proportion of two unrelated linear responses to the Equation (11) solves Schwartz’s differential equation [93].
(14)$$\left\{\varepsilon ,x\right\}=\frac{d}{dx}\left(\frac{\ddot{\varepsilon }}{\dot{\varepsilon }}\right)-\frac{1}{2}{\left(\frac{\ddot{\varepsilon }}{\dot{\varepsilon }}\right)}^{2}=2k_{0}^{2}$$
(15)$$\dot{\varepsilon }=\frac{d\varepsilon }{dx},\ddot{\varepsilon }=\frac{d^{2}\varepsilon }{dx^{2}}$$

The left side of the differential Equation (14) is unaffected in terms of homographic adjustments:(16)$$\varepsilon ↔\varepsilon^{\prime }=\frac{a\varepsilon +b}{c\varepsilon +d},\;\;a,b,c,d,\;e\in R$$

The group SL(2ℝ) is defined by relation (16), which considers all possible parameter settings. Therefore, every polymer–drug structural component possesses identical biunivocal correspondence with the transformations of the SL(2ℝ) group. The above allows the generation of a “personal” parameter identified as ε for every polymer–drug structural unit individually. In fact, serving as a support, the generic version of the solution of the differential Equation (14) can be selected, which is represented as
(17)$$\varepsilon^{\prime }=u+v\;tan\left(k_{0}x+\theta \right)$$

In such a context, through u, v, and θ effortlessly describe each polymer–drug structural unit. At this point, matching the step from solution (17) to the one from solution (13), the “personal” variable turns into
(18)$$\varepsilon^{\prime }=\frac{z+\bar{z}\varepsilon }{1+z},z=u+iv,\bar{z}=u-iv,\varepsilon \equiv e^{2i\left(k_{0}x+\theta \right)},\;i=\sqrt{-1}$$

The fact that solution (17) is also the result of the differential equation (14) indicates, by explaining the homographic transformations (16), the *SL*(2ℝ) group [31,85,86,87]:(19)$$z^{\prime }=\frac{az+b}{cz+d},\bar{z}=\frac{a\bar{z}+b}{c\bar{z}+d},k^{\prime }=\frac{c\bar{z}+d}{cz+d}k$$

Thus, group (19) works as “synchronization modes” among the various structural units of any polymer–drug complex. Both the amplitudes and phases of each of them clearly engage in this course of action in the sense that they are interdependent. This association indicates the situations listed below: (i) the phase shift of k is only changed with an amount correlated with the amplitude of the polymer–drug unit of structure at the point of switching between various structural subunits of any polymer–drug; (ii) the amplitude of the structural unit of every polymer–drug is impacted from a homographic standpoint; and (iii) the conventional “synchronization”, evidenced by a gap between the amplitudes and phases of the structural units of any polymer–drug, shall correspond to a distinctive scenario.

### 4.4. Simultaneities in Polymer-Drug Dynamics

The structure of group (19), i.e.,
(20)$$\left[L_{1},L_{2}\right]=L_{1},\left[L_{2},L_{3}\right]=L_{3},\left[L_{3},L_{1}\right]=-2L_{2}$$
is typical to *SL*(2ℝ). In relations (20), $L_{k},$ k=1,2,3 are the infinitesimal generators of the group. Because the group is simple transitive, the generators $L_{k},$ can be found as the components of the Cartan coframe from the relation [94]:(21)$$\begin{array}{cc} d\left(f\right)=\sum \frac{\partial f}{\partial x^{k}}dx^{k} & \\ & =\{\omega^{1}\left[z^{2}\frac{\partial }{\partial z}+{\bar{z}}^{2}\frac{\partial }{\partial \bar{z}}+\left(z-\bar{z}\right)k\frac{\partial }{\partial k}\right]+2\omega^{2}\left(z\frac{\partial }{\partial h}+\bar{z}\frac{\partial }{\partial \bar{z}}\right)\\ & +\omega^{3}\left(\frac{\partial }{\partial z}+\frac{\partial }{\partial \bar{z}}\right)\}\left(f\right) \end{array}$$

In Equation (21), $\omega^{k}$ are the elements of the Cartan coframe and may be obtained from the system [94].
(22)$$dz=\omega^{1}z^{2}+2\omega^{2}z+\omega^{3},d\bar{z}=\omega^{1}{\bar{z}}^{2}+2\omega^{2}\bar{z}+\omega^{3},dk=\omega^{1}k\left(z-\bar{z}\right)$$

Now, we can acquire both the infinitesimal generators and the Cartan coframe by identifying the right-hand side of relation (21) with the standard dot product of *SL*(2ℝ) algebra [94]:(23)$$\omega^{1}L_{3}+\omega^{3}L_{1}-2\omega^{2}L_{2}$$

It results in
(24)$$L_{1}=\frac{\partial }{\partial z}+\frac{\partial }{\partial \bar{z}},L_{2}=h\frac{\partial }{\partial z}+\bar{h}\frac{\partial }{\partial \bar{z}},L_{3}=z^{2}\frac{\partial }{\partial z}+{\bar{z}}^{2}\frac{\partial }{\partial \bar{z}}+\left(z-\bar{z}\right)k\frac{\partial }{\partial k}$$
and
(25)$$\omega^{1}=\frac{dk}{\left(z-\bar{z}\right)k},{2\omega }^{2}=\frac{dz-d\bar{z}}{z-\bar{z}}-\frac{z+\bar{z}}{z-\bar{z}}\frac{dk}{k},\omega^{3}=\frac{zdz-\bar{z}dz}{z-\bar{z}}+\frac{z\bar{z}dk}{\left(z-\bar{z}\right)k}$$

This allows us to reiterate the homographic transformation (16). Based on the previously given consequences of this adjustment, each structural component found in any polymer–drug could easily be identified employing homogenous coordinates $\left(a,b,c,d\right)$. The simultaneity requirement that applies to every polymer–drug’s structural units may be distinguished from a Riccati equation in pure differentials (referred to as a Riccati gauge).
(26)$$d\frac{a\varepsilon +b}{c\varepsilon +d}=0$$
which implies
(27)$$d\varepsilon =\omega^{1}\varepsilon^{2}+\omega^{2}\varepsilon +\omega^{3}$$
where $\omega^{1}$, $\omega^{2}$, and $\omega^{3}$ are the components of the Cartan coframe given through relations (25). Therefore, for the description of any polymer–drug dynamics as a succession of states of an ensemble of simultaneous structural units, as it were, it suffices to have three differentiable 1-forms, representing a coframe of *SL*(2ℝ) algebra. Consequently, a state of a polymer–drug in a given dynamics can be organized as a metric plane space, i.e., a Riemannian three-dimensional space. Accordingly, the geodesics of a Riemannian space are given by conservations of equations:(28)$$\omega^{1}=a^{1}d\tau ,\omega^{2}=a^{2}d\tau ,\omega^{3}=a^{3}d\tau$$
where $a^{1}$, $a^{2}$, and $a^{3}$ are uniform, and τ is the affine parameter of the geodesics. Along these geodesics relation (27) becomes a differential equation of Riccati type:(29)$$\frac{d\varepsilon }{d\tau }=a^{1}\varepsilon^{2}+2a^{2}\varepsilon +a^{3}$$

Permit us to analyze the subsequent situation of the previous differential equation:(30)$$A\frac{d\varepsilon }{d\tau }=P\left(\varepsilon \right)=\varepsilon^{2}-2B\varepsilon -AC$$
where we made the substitutions:(31)$$\frac{1}{a^{1}}=A,-2\frac{a^{2}}{a^{1}}=B,-\frac{a^{3}}{a^{1}}=AC$$

The roots of the polynomial $P\left(\varepsilon \right)$ can be referred to as
(32)$$\varepsilon_{1}=B+iA\Omega ,\varepsilon_{2}=B-iA\Omega ,\Omega^{2}=\frac{C}{A}-{\left(\frac{B}{A}\right)}^{2}$$
the change of variable
(33)$$z=\frac{\varepsilon -\varepsilon_{1}}{\varepsilon -\varepsilon_{2}}$$
transforms the differential equations (29) in
(34)$$\overset{´}{z}=2i\Omega z$$

The answer to the previous differential equation has the following form:(35)$$z\left(\tau \right)=z\left(0\right)e^{2i\Omega \tau }$$

Now, assuming the starting state *z*(0) is comfortably portrayed, it is feasible to obtain a general approach to the differential Equation (35) by inverting the transformation (33) with the following result:(36)$$\varepsilon =\frac{\varepsilon_{1}+re^{2i\Omega \left(\tau -\tau_{0}\right)}\varepsilon_{2}}{1+re^{2i\Omega \left(\tau -\tau_{0}\right)}}$$
where r and $\tau_{0}$ are two integration constants. Considering Formula (32), the outcome (36) may be expressed in the actual sense as
(37)$$z=B+A\Omega \left\{\frac{2rsin\left[2\Omega \left(\tau -\tau_{0}\right)\right]}{1+r^{2}+2rcos\left[2\Omega \left(\tau -\tau_{0}\right)\right]}+i\frac{1-r^{2}}{1+r^{2}+2rcos\left[2\Omega \left(\tau -\tau_{0}\right)\right]}\right\}$$

Therefore, the simultaneities in phase and amplitude of the polymer–drug structural units imply group invariances of *SL*(2ℝ) type. Then, period doubling, damped fluctuations, quasi-periodicity, intermittence, etc., emerge as natural behaviors in the polymer–drug complex dynamics (see Figure 2, Figure 3, Figure 4 and Figure 5 for r=0.5 and Real $\left[\left(z-B\right)/A\right]\equiv F\left(\omega ,t\right)$ ≡ Amplitude at various scale resolutions, given by means of the maximum value of $\Omega_{max}$).

As may be noticed in Figure 2, Figure 3, Figure 4 and Figure 5, the natural transition of a polymer–drug is to progress from a normal period doubling state towards damped fluctuating and a strong modulated dynamic. The polymer–drug never achieves a chaotic state, although it does make progress toward it. The polymer–drug leaps instantly into a doubling period state, repeating the event presented above.

The development of the polymer–drug system is continuing to be studied by increasing the control parameter. To achieve this objective, the polymer–drug complex’s response is examined across a restricted spectrum of values. It is observed (see Figure 2, Figure 3, Figure 4 and Figure 5) that the polymer–drug dynamics starts from a double period state and progresses to a reduced fluctuating state before advancing toward a quasi-chaotic state that is ultimately not reached. The existence of extra oscillation frequencies serves as proof of the changeover. Despite the fact that the frequency output of the polymer–drug structure complex is regular, the amplitude grows approximately linearly as the control variable improves in value. The bifurcation map is presented in Figure 6, and it is a representation of the above-stipulated situation: the initial steady state of the polymer–drug complex dynamics and its tendency toward a chaotic one ($\Omega_{max}=2,2.5,3\ldots$) without being able to achieve it.

### 4.5. Some Implications of the Multifractal Madelung Scenario in Drug Delivery Dynamics

The functionality of the multifractal Schrödinger scenario in the definition of drug delivery dynamics implies synchronizations and simultaneities of the polymer–drug structural unit dynamics. These are transmitted both at differentiable and non-differentiable resolution scales and also at the differentiable–non-differentiable scale transition. Thus, it is imperative that the differentiable and non-differentiable velocities satisfy the condition:(38)$$v_{D}^{i}=-v_{F}^{i}$$

In such a conjecture, the multifractal hydrodynamics equations system (see Equations (7) and (8)) reduces to the multifractal diffusion equation:(39)$$\partial_{t}\rho =\sigma \partial_{l}\partial^{l}\rho$$
where σ defines the multifractal diffusion coefficient.
(40)$$\sigma =\sigma {\left(dt\right)}^{\left[\frac{2}{f\left(\alpha \right)}\right]-1}$$

Such a relation can describe drug release dynamics of Fickian and non-Fickian types in accordance with standard drug release models [31,88,89,90,91,92,95].

In terms of the potential function of polymer-based drug delivery systems in clinical practice, it is worth noting that various research works have been conducted thus far. As a result, several studies emphasize their importance not only in obtaining a diagnosis but also in determining the main tumor site and, indirectly, staging. Furthermore, the response to the therapy may be determined by utilizing these systems, taking into account their capacity to provide targeted treatment under changeable environmental circumstances [96,97,98].

In the context of their role as nanocarriers, polymers exhibit an effective capacity for dispensing the material carried in the targeted environment, a boost in its internalization in the targeted cells, and low toxic consequences for other cells, which is why they are successfully used in research in this setting [99,100].

Given the ethical concerns of drug delivery system research, particularly with regard to the topic of cancer therapy, it is critical to address the potential health hazards associated with its usage. Thus, while it intends to target neoplastic cells, their use may have the opposite effect by causing genetic modifications associated with neoplastic cell proliferation while also emphasizing the negative environmental effects of these substances [101]. As a result, access to this sort of therapy and obtaining informed consent must be strictly monitored, which is also true with regard to the privacy of these individuals.

## 5. Conclusions

Regarding the primary concepts that emerged from the findings of the present paper:(i)A unitary model for describing disperse dynamics in polymer–drug systems has been suggested, considering that such a system can potentially be perceived as a multifractal mathematical object.(ii)Using this theory, the polymer–drug dynamics can be referred to by employing the Scale Relativity Theory in the monofractal or the multifractal case.(iii)This approach allows the characterization of polymer–drug dynamics through fractal/multifractal curves. Since these graphical representations display the self-similarity propriety, it can be stated that the description of these dynamics implies holographic-type behaviors; therefore, a holographic model for polymer–drug dynamics can be developed.(iv)The use of a holographic model for describing drug release dynamics can be reduced to two scenarios, namely a Schrödinger-type scenario and a Madelung-type scenario.(v)In the Schrödinger-type scenario, the synchronization of any polymer–drug structural unit’s dynamics implies the *SL*(2ℝ) group, while the simultaneity of the same dynamics through a Riccati-type gauge implies various release modes, “mimed” through several behaviors, such as period doubling (which can be associated to the swelling of the polymeric complex), damped oscillations regimes (which can be associated to surface release mechanisms), self-modulated regimes (which can be associated to in-depth release mechanisms), or chaotic regimes (which can be associated to polymer degrading release mechanisms).(vi)In the Madelung-type scenario, the same processes that are “functional” in the Schrödinger-type scenario (synchronization and simultaneity) imply dynamics described through multifractal diffusion equations. By these equations, Fickian, non-Fickian, or Fickian–non-Fickian transitory behaviors can be substantiated in drug release dynamics.

## Figures and Tables

**Figure 1 pharmaceuticals-17-00541-f001:**
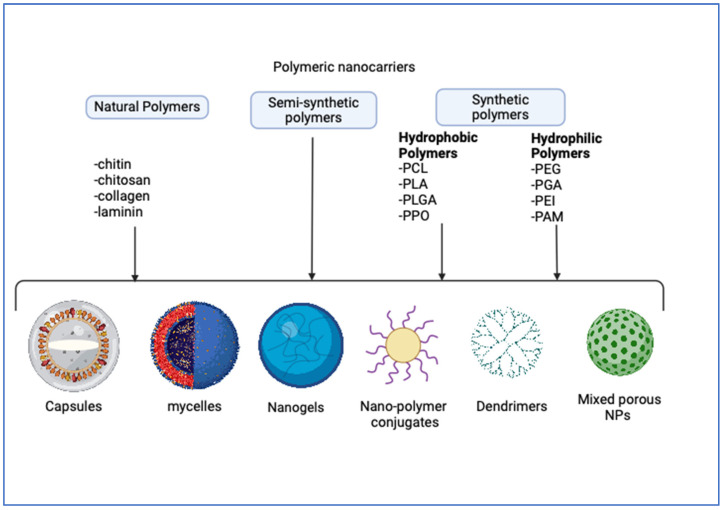
Various polymer nanocarrier-based pharmaceutical delivery methods for treatment of tumors (Biorender.com).

**Figure 2 pharmaceuticals-17-00541-f002:**
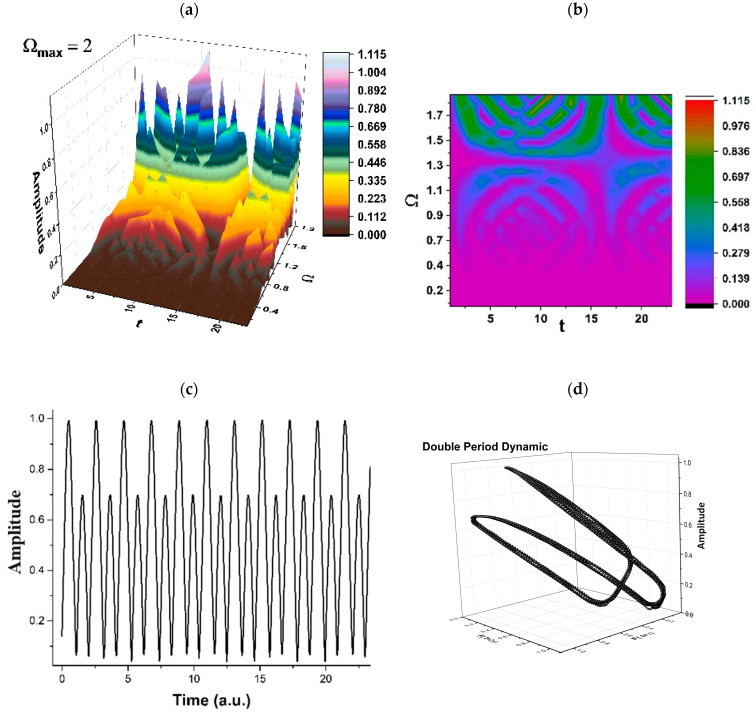
(**a**–**d**). Period doubling regimes in polymer–drug dynamics: (**a**) 3D representation; (**b**) 2D representation; (**c**) time series; (**d**) reconstructed attractor for scale resolution given by $\Omega_{max}=2$.

**Figure 3 pharmaceuticals-17-00541-f003:**
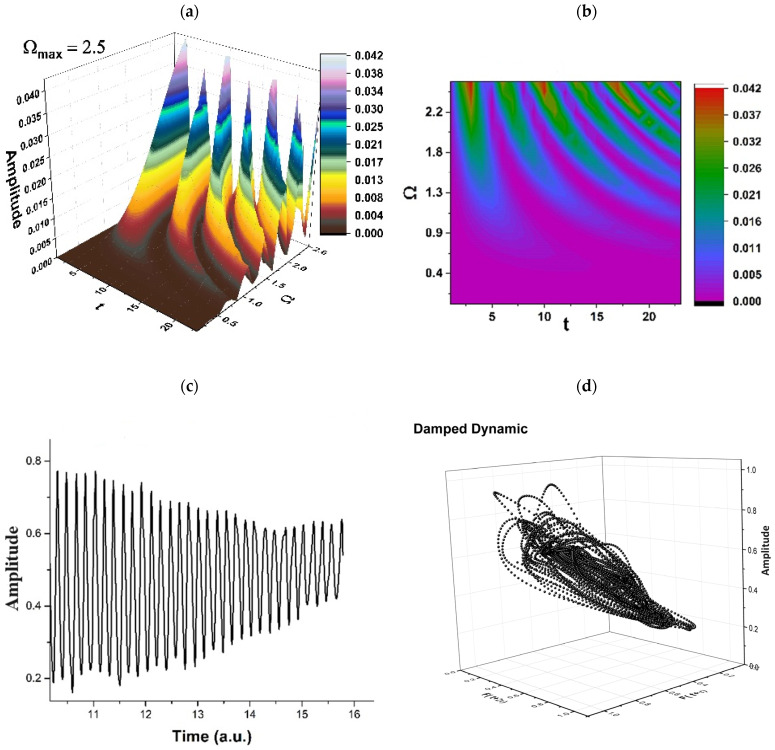
(**a**–**d**). Damped oscillation regimes in polymer–drug dynamics: (**a**) 3D representation; (**b**) 2D representation; (**c**) time series; (**d**) reconstructed attractor for scale resolution given by $\Omega_{max}=2.5$.

**Figure 4 pharmaceuticals-17-00541-f004:**
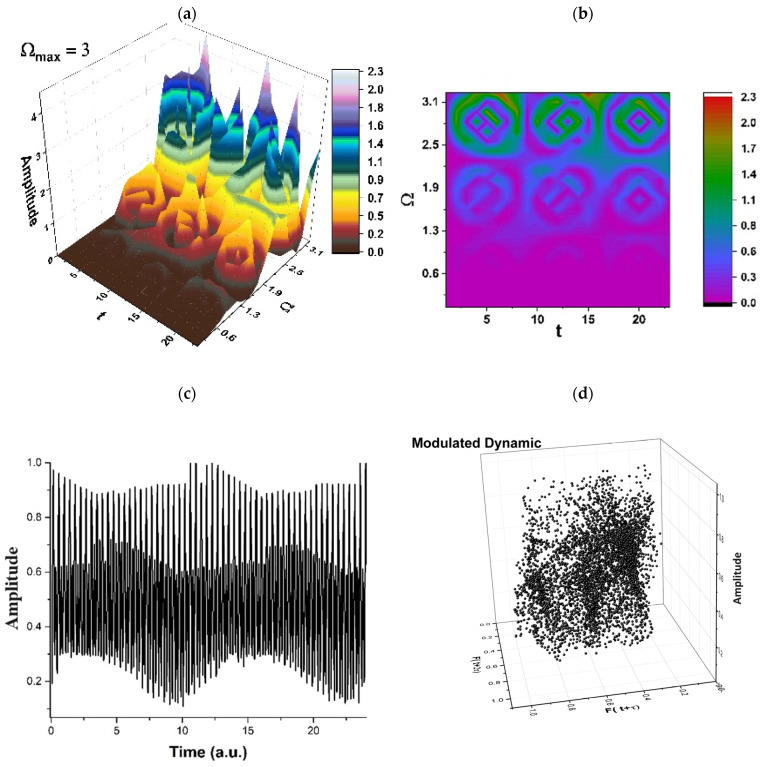
(**a**–**d**). Modulated regimes in polymer–drug dynamics: (**a**) 3D representation; (**b**) 2D representation; (**c**) time series; (**d**) reconstructed attractor for scale resolution given by $\Omega_{max}=3$.

**Figure 5 pharmaceuticals-17-00541-f005:**
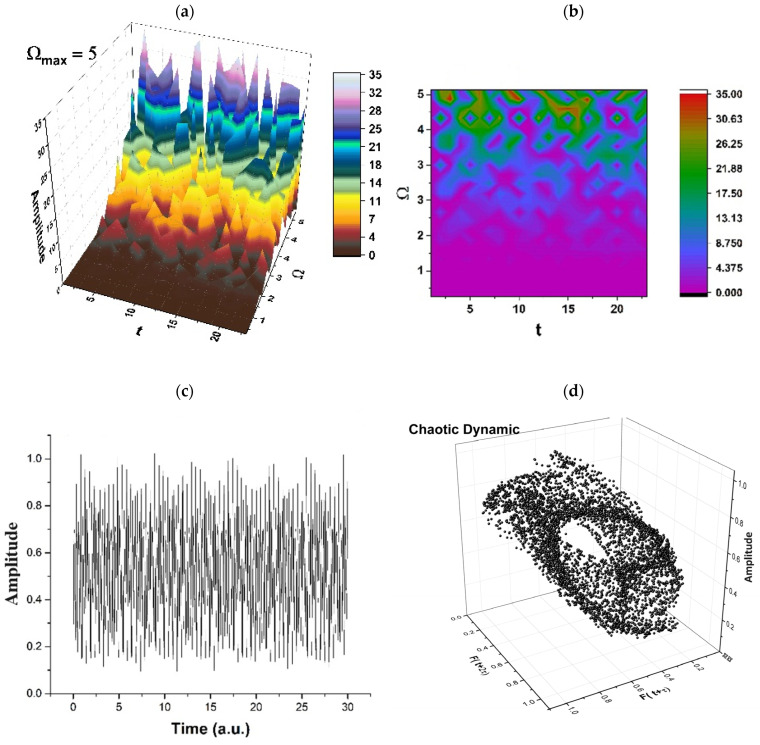
(**a**–**d**). Chaotic regimes in polymer–drug dynamics: (**a**) 3D representation; (**b**) 2D representation; (**c**) time series; (**d**) reconstructed attractor for scale resolution given by $\Omega_{max}=5$.

**Figure 6 pharmaceuticals-17-00541-f006:**
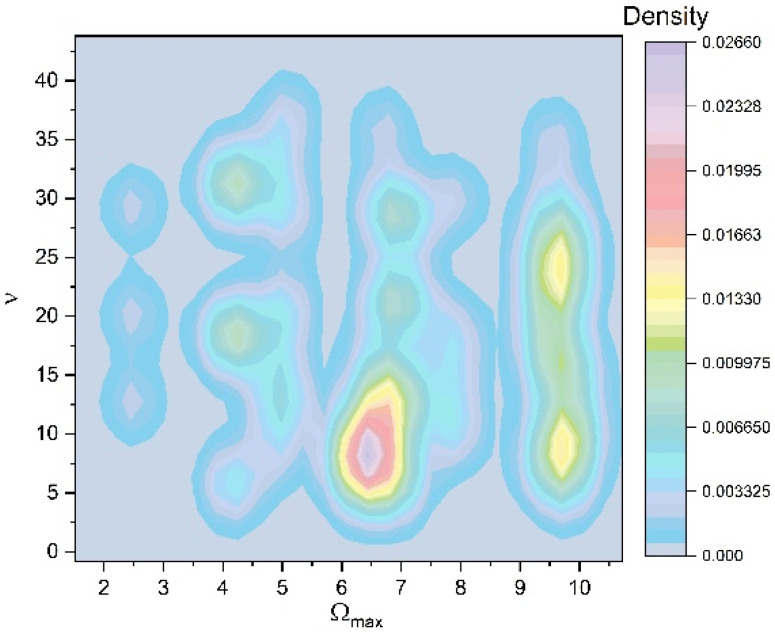
Frequency of the polymer–drug structural units as a function of a scale resolution chosen by $\Omega_{max}$. Bifurcation map.

## Data Availability

Data is contained within the article.

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
