# Peer review of "A Holographic-Type Model in the Description of Polymer–Drug Delivery Processes"

_pharmaceuticals, 2024, doi:10.3390/ph17040541_

Round 1
Reviewer 1 Report
Comments and Suggestions for Authors
Major Revision
This is an interesting paper that focuses on the description of polymer-drug delivery model. The content described in this manuscript definitely falls into the scope of this journal. Therefore, I recommend this work could be published after the major revision.
1) No graphical abstract has been provided.
2) The introduction part is not supported by enough references. Why the authors prepared such functionalized materials comparing with the recently reported different materials such as Metal-organic frameworks? Author should cite some relevant papers. For example; https://doi.org/10.1016/j.molliq.2019.112060
3) The more elaborated mechanisms or explanations (in section 3) are required to explain.
4) The effect of pH is very crucial in drug delivery therefore author should consider various pH to explain the model.
5) Author should cite new papers. For example ref. 49 and 55 was too old.
Author Response
Thank you for your review. The answers to the comments are the following:
1.A graphical abstract has been provided.
2.We added some new info in this regard
3.We added an inttroductory part in Section 3, in order to better explain the presented mechanisms.
4.In the updated part of Section 3, we presented how our model can incorporate various pH, indicating two relevant refecences
5.We have replaced refs. 49 and 55 with more recent ones.

Reviewer 2 Report
Comments and Suggestions for Authors
This paper provides a comprehensive overview of polymer-based drug delivery systems and their role in cancer treatment, covering various types of polymers, nanoparticles, and drug release dynamics. The introduction sets a solid foundation, but the complexity of the mathematical models introduced later may challenge some readers. Further clarification on the basis of the following points and simplification would enhance the overall impact of this paper:
- Could the introduction provide a clearer outline of the paper's structure to guide readers through the complex subject matter?
- How can the paper simplify the discussion of mathematical models to make it more accessible to a broader audience, particularly those without a strong background in mathematics?
- Are there specific examples or case studies that could be included to illustrate the practical application of the discussed drug delivery systems?
- Could the paper discuss any potential limitations or challenges associated with the implementation of polymer-based drug delivery systems in clinical settings?
- How might the paper address any potential ethical considerations or safety concerns related to the use of nanoparticles in drug delivery systems, particularly in the context of cancer treatment?
Author Response
Thank you for your review. The answers to the comments are:
1.We added a Results section, after the Introduction, in order to comply with the Journal s article structure. In this section, we provided a clearer outline of the paper s structure and results.
2.We added an introductory part in Section 3, in order to better explain the presented mechanisms.
3.We added the examples in the discussion topic.
4.We added some info about this topic.
5.There is little information about this aspect in the literature. The topic can be found in Conclusions section

Reviewer 3 Report
Comments and Suggestions for Authors
The research work by Irina Nica and team on “A holographic-type model in the description of polymer-drug” proposed a model of drug release dynamics (via scale relativity theory) based on an assumption the polymer-drug system can be assimilated to a multifractal mathematical object. They demonstrated the hypothesis of the use of holographic image reconstruction methods for monitoring drug release efficiency. The proposed model looks applicable and useful, however some of my comments are as follows:
1. Introduction is too long, and much data may not be relevant. Only discuss about need of a model and how it can overcome old model problems or inaccuracies.
2. Correlation of this theory and imaging technique with available simple techniques is required.
3. Authors should verify the model by incorporating the hypothetical data and outcomes must be compared with other models.
4. Discuss about limitations of this model, like time consumption, accuracy, predictability etc
5. Will this model apply to all, dissolution types as we all diffusion-type polymers?
6. Work lacks the discussion part, as it is briefly discussed along with a conclusion.
7. No methodology was found in this work.
Author Response
Thank you for your revision. The answers to the comments are:
1.We have made the required changes
2. We added a Discussion Section, in which we discuss how the standard imaging methods can be correlated with our model.
3.We added an introductory part in Section 3, in order to better explain the presented mechanisms. In this part, we explain how different release mechanisms can be correlated with the various factors that influence drug release. Our model is an original one, because it stands as a unitary approach to drug release processes, contrary to semi-empirical laws employed in classical models.
4.We added a Discussions Section , in which we discuss about the limitations of our model.
5.The goal of this paper was not to find a universal applicable model.
6.We added a Discussion Section
7. Our work presents a theoretical model. We reorganized the sections presenting this model in a Methods Section, in order to comply with the Journal s structure

Round 2
Reviewer 1 Report
Comments and Suggestions for Authors
Thanks for your effort even though effect of pH on delivery model is not much clear.
Reviewer 3 Report
Comments and Suggestions for Authors
I am satisfied with the revised version and justifications.